**Data Availability Statement:** All relevant data are within the paper. The full data correspond to Table 1 included within the paper.

# Spatial epidemiological study of the distribution, clustering, and risk factors associated with early COVID-19 mortality in Mexico

**Ricardo Ramírez-Aldana** [ID] *, **Juan Carlos Gomez-Verjan, Omar Yaxmehen Bello-Chavolla** [ID], **Carmen García-Peña** [ID]

Research Division, Instituto Nacional de Geriatria, Mexico City, Mexico

* ricardoramirezaldana@gmail.com

## Abstract

COVID-19 is a respiratory disease caused by SARS-CoV-2, which has significantly impacted economic and public healthcare systems worldwide. SARS-CoV-2 is highly lethal in older adults (>65 years old) and in cases with underlying medical conditions, including chronic respiratory diseases, immunosuppression, and cardio-metabolic diseases, including severe obesity, diabetes, and hypertension. The course of the COVID-19 pandemic in Mexico has led to many fatal cases in younger patients attributable to cardio-metabolic conditions. Thus, in the present study, we aimed to perform an early spatial epidemiological analysis for the COVID-19 outbreak in Mexico. Firstly, to evaluate how mortality risk from COVID-19 among tested individuals (MRt) is geographically distributed and secondly, to analyze the association of spatial predictors of MRt across different states in Mexico, controlling for the severity of the disease. Among health-related variables, diabetes and obesity were positively associated with COVID-19 fatality. When analyzing Mexico as a whole, we identified that both the percentages of external and internal migration had positive associations with early COVID-19 mortality risk with external migration having the second-highest positive association. As an indirect measure of urbanicity, population density, and overcrowding in households, the physicians-to-population ratio has the highest positive association with MRt. In contrast, the percentage of individuals in the age group between 10 to 39 years had a negative association with MRt. Geographically, Quintana Roo, Baja California, Chihuahua, and Tabasco (until April 2020) had higher MRt and standardized mortality ratios, suggesting that risks in these states were above what was nationally expected. Additionally, the strength of the association between some spatial predictors and the COVID-19 fatality risk varied by zone.

**Funding:** The publication of this paper was supported by a grant from the Secretaría de Educación, Ciencia, Tecnología e Innovación de la Ciudad de México CM-SECTEI/200/2020 "Red Colaborativa de Investigación Traslacional para el Envejecimiento Saludable de la Ciudad de México (RECITES). The funders had no role in study design, data collection and analysis, decision to publish, or preparation of the manuscript.

**Competing interests:** The authors have declared that no competing interests exist.

## Introduction

COVID-19 is a respiratory disease caused by SARS-CoV-2 (severe acute respiratory syndrome coronavirus 2), which has caused almost twenty million cases worldwide and caused 790,000 deaths as of August 20[th], 2020 [1]. SARS-CoV-2 is a highly contagious RNA virus from the Coronaviridae family, with a small genome of about 30,000 nucleotides closely related to the bat coronavirus (RaTG13) [2]. Given its global spread, there is an urgent need for scientific and health systems around the world to understand the epidemiology, pathogenicity, and mechanisms of immunological defences to develop possible therapeutic and public health alternatives to fight against one of the most outstanding threats to public health since Spanish Influenza over a hundred years ago [3].

According to the World Health Organization (WHO), the groups most susceptible to acquire infection and develop adverse outcomes are people with underlying medical conditions and older adults (>65 years old), particularly those living in nursing homes. Medical conditions which have been associated with increased susceptibility for adverse outcomes related to SARS-CoV-2 infection include chronic obstructive pulmonary disease (COPD), chronic kidney disease (CKD), cardiovascular diseases, liver diseases, moderate asthma, immunosuppression (HIV/AIDS, bone marrow transplantation, cancer treatment, and genetic immune deficiencies), and particularly, severe obesity, diabetes, and hypertension [4]. These associations may be related to the strong link between pro-inflammatory cytokines in response to infection and the pathogenesis of SARS-CoV-2, which can be seen in pneumonia patients with severe COVID-19 disease exhibiting systemic hyper-inflammation known as cytokine storm or as a secondary hemophagocytic lymphohistiocytosis [5].

According to the Organisation for Economic Co-operation and Development (OECD), in the last ten years, the countries with the highest prevalence of cardio-metabolic conditions that are now linked to increased risk of severe COVID-19, including obesity, type 2 diabetes, and hypertension, are Mexico and the United States of America (USA), when considering adults from 15 to 74 years old [6]. Thus, considering that the population pyramid is becoming more rectangular through time, indicating an aging population structure, the high rates of diabetes and obesity in Mexico are likely to lead to higher mortality rates attributable to COVID-19 even in younger populations [7].

In response to SARS-CoV-2 spread worldwide, mobility restrictions have been imposed to reduce community-level transmission; nevertheless, the early influence of human mobility, particularly internal and external migration, has primarily driven the spread of COVID-19 [8]. When considering the high transmissibility of SARS-CoV-2, human mobility gains relevance in the early stages of spread, in which people travelling from other countries can drive increasing transmission rates [9]. SARS-CoV-2 spread related to human mobility is relevant, particularly considering that the risk of transmission and adverse outcomes are related to inequalities and suboptimal socio-economic conditions [10]. In this sense, the risk of becoming infected and dying might increase in areas without optimum socio-economic conditions, such as spatial units in which people live in overcrowded households or without access to potable water or drainage.

Spatial analyses allow us to understand how mortality risks are distributed throughout a territory, detect spatial clusters, and measure how the effects of variables associated with this risk vary within any given territory. In this sense, several examples of geo-epidemiological studies in autoinflammatory diseases, specific syndromes, infectious diseases, and other conditions, have helped develop public health policies and knowledge of disease spread [11–13]. In the present work, we performed spatial analyses to measure spatial relationships corresponding to the geographical phenomenon of the COVID-19 outbreak in Mexico and its associated fatality

risks. We used statistical methods, including spatial clustering through local indicators of spatial autocorrelation and generalized geographically weighted regression to accomplish these objectives. Additionally, we aimed to identify spatial units or regions in Mexico that should be examined in more detail to better understand the propagation of the disease and its associated fatality risk in the early stages of COVID-19 spread.

## Material and methods

### Data sources

We obtained state-level variables concerning the 32 states of Mexico (Tables 1 and 2). Individual-level variables were obtained from Mexico's epidemiological surveillance entity (Direccion General de Vigilancia Epidemiologica, Secretaria de Salud), the data set corresponding to observations until April 21$^{st}$ 2020 [14]. These variables concern health and very general socio-economic information associated with people who were suspected of having COVID-19 in Mexico (people who by their concerns were suspected of having the disease, presenting symptoms, or who were in contact with someone with COVID-19) and underwent real-time RT-PCR for SARS-CoV-2 confirmation. Available variables include the presence of diabetes, obesity, chronic renal problems (CKD), chronic obstructive pulmonary disease (COPD), pregnancy, hypertension, immunosuppression, cardiovascular disease, pneumonia, as well as age, and whether the patient was hospitalized, admitted to an intensive care unit (ICU), or required intubation. We grouped age into groups, as explained in the model selection process below, and we finally used three age groups for subsequent analyses: 10–39, 40–69, and 70 years old and over. We also computed the risk of death due to the disease on individuals who were tested for SARS-CoV-2, or mortality risk from COVID-19 among tested individuals (MRt), by considering as a death that which was recorded after having a positive test (the information concerning positive tests and death is available in the epidemiological data set), a method consistent with the official numbers. We aggregated all variables at a state level as counts and used them as relative frequencies (risks for the health variables and proportion of individuals for each age group) in all analyses.

Additional variables concerning socio-demographic, economic, mobility, and climatic features were obtained at a state-level. In terms of socio-demographic variables, from the National Institute of Geography and Statistics (INEGI), we extracted information concerning population density in various contexts: density as the number of people per $km^2$ in 2015 and the proportion of people in a household living in a crowded place in 2017; literacy rate of population aged ≥15 years in 2015; people settled in rural areas in 2010 (%) (a location was considered as rural when there were <2,500 habitants); and the number of physicians available for every 1000 people in 2015, which was obtained from the National Institute of Public Health (INSP). In terms of economic variables, we obtained from INEGI the state contribution to gross domestic product (GDP) in 2018; however, we used the multiplicative effect between a dummy variable concerning the states with the biggest cities in Mexico and GDP, in which the linearity assumption with the transformed response was improved. We also obtained information concerning people living in poverty in 2018 (%), as it is defined and calculated by CONEVAL according to a multidimensional index obtained from *per capita* income and an index of social deprivation [15]. In terms of mobility, we extracted from INEGI information concerning internal migration, as the rate of people aged five years and over living in another state five years before 2014, and external migration, as the rate of people aged ≥5 years living in another country five years before 2014, both proxies of internal and external mobility, respectively. Concerning mobility, we also used the number of flights in 2019 by state, which we calculated from information associated with the number of flights by the airport in Mexico (Ministry of

**Table 1. Features extracted for all the analyses by state used to predict the mortality risk from COVID-19 among tested individuals in Mexico.**

| State | Deaths caused by COVID-19[*] | Number of tested individuals[a,*] | Rural 2006 (%)[**] | Poverty 2018 (%)[***] | Density 2015[+] | Literacy 2015 (%)[+] | GDP 2018 (%)[+] | Temperature March 2020 (°C)[++] | Internal migration 2014 (%)[+] | External migration 2014 (%)[+] | Physicians by 1000 people 2015[+++] | Overcrowding 2017 (%)[+] |
|---|---|---|---|---|---|---|---|---|---|---|---|---|
| AGUASCALIENTES | 2 | 1070 | 18.84 | 26.18 | 233.70 | 97.00 | 1.31 | 19.30 | 3.60 | 0.70 | 1.33 | 3.90 |
| BAJA CALIFORNIA | 100 | 2353 | 6.99 | 23.26 | 46.40 | 97.60 | 3.14 | 14.40 | 3.80 | 1.90 | 1.05 | 6.40 |
| BAJA CALIFORNIA SUR | 9 | 978 | 15.22 | 18.07 | 9.60 | 96.80 | 0.98 | 19.80 | 8.20 | 0.50 | 1.50 | 10.40 |
| CAMPECHE | 4 | 229 | 26.00 | 46.25 | 15.60 | 92.90 | 2.98 | 28.00 | 4.10 | 0.20 | 1.16 | 16.50 |
| COAHUILA DE ZARAGOZA | 23 | 2472 | 9.95 | 22.49 | 19.50 | 97.10 | 3.50 | 22.10 | 2.10 | 0.40 | 1.44 | 4.50 |
| COLIMA | 2 | 133 | 12.41 | 30.87 | 126.40 | 95.60 | 0.61 | 26.60 | 5.60 | 1.50 | 1.56 | 10.70 |
| CHIAPAS | 4 | 440 | 52.26 | 76.41 | 71.20 | 84.20 | 1.56 | 25.60 | 1.50 | 0.50 | 0.53 | 20.50 |
| CHIHUAHUA | 36 | 705 | 15.51 | 26.28 | 14.40 | 95.00 | 3.20 | 16.70 | 1.20 | 1.20 | 1.11 | 5.60 |
| CIUDAD DE MÉXICO | 224 | 10183 | 0.35 | 30.55 | 5967.30 | 97.70 | 17.64 | 19.40 | 2.10 | 0.40 | 2.99 | 7.10 |
| DURANGO | 4 | 407 | 32.77 | 37.35 | 14.20 | 96.20 | 1.13 | 18.40 | 2.60 | 0.80 | 1.17 | 5.40 |
| GUANAJUATO | 9 | 2482 | 30.29 | 43.38 | 191.30 | 93.00 | 4.11 | 20.60 | 1.50 | 1.00 | 0.84 | 5.50 |
| GUERRERO | 16 | 583 | 42.44 | 66.47 | 55.60 | 85.50 | 1.37 | 26.20 | 1.90 | 1.10 | 0.64 | 27.60 |
| HIDALGO | 16 | 608 | 47.69 | 43.85 | 137.30 | 91.00 | 1.54 | 20.50 | 3.90 | 0.90 | 0.66 | 8.20 |
| JALISCO | 15 | 3440 | 13.85 | 28.43 | 99.80 | 95.80 | 6.91 | 22.40 | 2.20 | 1.10 | 1.26 | 5.30 |
| MÉXICO | 71 | 5759 | 12.74 | 42.72 | 724.20 | 95.80 | 8.93 | 16.30 | 2.80 | 0.40 | 0.81 | 8.60 |
| MICHOACÁN | 19 | 1124 | 32.06 | 46.04 | 78.20 | 90.80 | 2.43 | 21.70 | 1.70 | 1.20 | 0.72 | 10.10 |
| MORELOS | 14 | 525 | 13.91 | 50.82 | 390.20 | 94.30 | 1.13 | 24.70 | 3.30 | 0.90 | 1.04 | 10.80 |
| NAYARIT | 5 | 299 | 33.48 | 34.77 | 42.40 | 94.30 | 0.68 | 24.70 | 4.30 | 1.40 | 1.06 | 8.10 |
| NUEVO LEÓN | 6 | 4482 | 5.64 | 14.53 | 79.80 | 97.40 | 7.47 | 22.90 | 2.70 | 0.40 | 1.30 | 3.10 |
| OAXACA | 7 | 522 | 52.89 | 66.35 | 42.30 | 84.20 | 1.48 | 24.60 | 2.00 | 1.20 | 0.61 | 16.90 |
| PUEBLA | 53 | 1651 | 29.38 | 58.92 | 179.80 | 90.80 | 3.39 | 20.00 | 2.30 | 0.60 | 0.83 | 9.70 |
| QUERÉTARO | 6 | 630 | 30.13 | 27.58 | 174.40 | 94.70 | 2.33 | 22.00 | 5.60 | 0.90 | 0.85 | 6.90 |
| QUINTANA ROO | 47 | 982 | 14.42 | 27.57 | 33.60 | 95.50 | 1.63 | 26.80 | 8.00 | 0.70 | 0.96 | 17.50 |
| SAN LUIS POTOSÍ | 5 | 1167 | 37.31 | 43.40 | 44.50 | 92.90 | 2.11 | 24.60 | 2.70 | 0.80 | 0.85 | 6.20 |
| SINALOA | 52 | 1580 | 29.17 | 30.88 | 51.70 | 95.20 | 2.23 | 22.90 | 2.50 | 0.70 | 1.16 | 9.20 |
| SONORA | 13 | 869 | 14.23 | 28.19 | 15.90 | 96.70 | 3.26 | 17.70 | 2.90 | 1.10 | 1.39 | 7.00 |
| TABASCO | 55 | 1266 | 44.98 | 53.65 | 96.90 | 93.80 | 2.61 | 27.60 | 2.30 | 0.30 | 0.92 | 12.00 |
| TAMAULIPAS | 7 | 1210 | 12.74 | 35.07 | 42.90 | 96.00 | 2.82 | 25.20 | 1.80 | 0.30 | 1.20 | 6.90 |
| TLAXCALA | 5 | 587 | 21.77 | 48.38 | 318.40 | 95.20 | 0.56 | 16.30 | 3.40 | 0.60 | 0.81 | 9.50 |
| VERACRUZ | 13 | 1520 | 39.33 | 61.78 | 113.00 | 89.80 | 4.59 | 23.70 | 3.00 | 0.60 | 0.87 | 12.60 |
| YUCATÁN | 12 | 959 | 17.01 | 40.80 | 53.10 | 91.90 | 1.46 | 26.90 | 3.60 | 0.40 | 1.39 | 13.90 |
| ZACATECAS | 3 | 419 | 42.73 | 46.76 | 21.00 | 94.90 | 0.88 | 19.10 | 2.10 | 1.40 | 0.89 | 5.00 |

* Dirección General de Epidemiología, Secretaría de Salud (General Direction of Epidemiology, Health Ministry).

** Instituto Nacional de Salud Pública, Encuesta Nacional de la Dinámica Demográfica (National Public Health Institute, National Survey of Demographic Dynamic).

*** Estimations obtained by CONEVAL (Consejo Nacional de Evaluación de la Política de Desarrollo Social, National Council for the Evaluation of Social Development Policy) based on the ENIGH (National Survey of Household Income and Expenditure) 2008, 2010, 2012, 2014, 2016, and 2018.

+ Instituto Nacional de Estadística y Geografía (National Institute of Statistics and Geography).

++ Consejo Nacional del Agua (CONAGUA) (National Water Comission).

+++ Instituto Nacional de Salud Pública (INSP) (The National Institute of Public Health of Mexico).

a The natural log of this number was used as the offset in all Poisson regression models.

Abbreviations: GDP, Gross Domestic Product; COPD, Chronic Obstructive Pulmonary Disease; and ICU, Intensive Care Unit.

**Table 2.**

| State | Diabetes (%)* | Obesity (%)* | Chronic renal problems (%)* | COPD (%)* | Pregnancy (%)* | Hyper-tension (%)* | Immuno-deficiency (%)* | Cardio-vascular (%)* | Pneumonia (%)* | ICU (%)* | Intubated (%)* | Hospitalized (%)* | Age group 10-39(%)* | Age group 40-69(%)* | Age group 70 years and over(%)* |
|---|---|---|---|---|---|---|---|---|---|---|---|---|---|---|---|
| AGUASCALIENTES | 7.22 | 14.20 | 1.95 | 1.83 | 0.57 | 13.40 | 1.83 | 2.86 | 7.45 | 0.46 | 0.92 | 12.26 | 54.30 | 33.33 | 4.58 |
| BAJA CALIFORNIA | 12.42 | 18.06 | 1.57 | 1.71 | 1.07 | 22.48 | 2.36 | 3.85 | 21.48 | 1.21 | 2.78 | 27.19 | 49.18 | 43.68 | 3.85 |
| BAJA CALIFORNIA SUR | 11.35 | 10.76 | 0.98 | 0.98 | 2.54 | 14.68 | 2.15 | 3.13 | 13.50 | 1.57 | 1.37 | 17.61 | 53.82 | 35.23 | 4.89 |
| CAMPECHE | 11.11 | 16.24 | 0.00 | 0.85 | 0.00 | 12.82 | 2.14 | 2.14 | 16.67 | 2.14 | 1.28 | 26.92 | 46.58 | 45.30 | 5.56 |
| COAHUILA DE ZARAGOZA | 11.96 | 16.08 | 2.02 | 1.85 | 0.76 | 16.04 | 2.60 | 2.98 | 8.94 | 1.85 | 0.67 | 14.48 | 47.40 | 44.25 | 5.58 |
| COLIMA | 14.39 | 19.42 | 5.04 | 2.88 | 2.16 | 15.83 | 2.16 | 3.60 | 25.90 | 2.88 | 2.88 | 33.09 | 52.52 | 34.53 | 7.19 |
| CHIAPAS | 11.84 | 13.05 | 1.97 | 2.88 | 1.67 | 15.63 | 2.28 | 2.73 | 17.30 | 3.64 | 3.03 | 27.31 | 49.47 | 39.15 | 4.86 |
| CHIHUAHUA | 15.38 | 13.50 | 2.83 | 4.71 | 1.41 | 21.82 | 3.30 | 5.18 | 33.44 | 9.73 | 3.92 | 46.00 | 46.00 | 40.82 | 7.85 |
| CIUDAD DE MÉXICO | 10.80 | 15.51 | 1.66 | 1.87 | 0.62 | 14.97 | 2.69 | 3.05 | 14.67 | 2.04 | 2.11 | 20.36 | 44.62 | 47.93 | 4.77 |
| DURANGO | 15.61 | 13.62 | 2.99 | 4.98 | 0.83 | 24.92 | 4.49 | 5.98 | 19.77 | 2.33 | 2.16 | 33.39 | 43.36 | 39.20 | 13.79 |
| GUANAJUATO | 10.13 | 12.00 | 1.91 | 3.94 | 1.40 | 14.46 | 2.76 | 2.67 | 10.43 | 1.70 | 1.53 | 17.34 | 51.00 | 34.72 | 6.87 |
| GUERRERO | 15.46 | 17.53 | 2.80 | 3.22 | 1.24 | 17.63 | 1.76 | 3.73 | 26.35 | 5.39 | 4.25 | 33.09 | 43.15 | 44.81 | 7.57 |
| HIDALGO | 18.49 | 15.98 | 4.14 | 4.73 | 1.48 | 19.97 | 4.44 | 2.07 | 21.89 | 1.63 | 2.96 | 31.95 | 42.75 | 44.23 | 9.17 |
| JALISCO | 10.88 | 13.43 | 3.53 | 3.63 | 1.47 | 16.56 | 2.89 | 3.84 | 12.44 | 1.87 | 1.26 | 20.40 | 49.12 | 37.67 | 7.96 |
| MÉXICO | 12.90 | 15.37 | 2.36 | 2.36 | 0.84 | 15.14 | 2.75 | 3.08 | 24.32 | 2.86 | 3.25 | 33.16 | 45.73 | 44.56 | 5.94 |
| MICHOACÁN | 13.54 | 16.00 | 2.75 | 4.27 | 1.38 | 19.41 | 2.46 | 4.34 | 22.23 | 2.17 | 2.46 | 29.62 | 42.00 | 41.20 | 11.37 |
| MORELOS | 14.40 | 14.79 | 2.92 | 4.09 | 1.17 | 17.90 | 2.72 | 2.53 | 24.71 | 4.09 | 3.50 | 34.82 | 43.77 | 42.22 | 9.73 |
| NAYARIT | 16.19 | 16.45 | 4.18 | 4.70 | 0.26 | 23.76 | 3.39 | 4.44 | 20.89 | 2.35 | 2.35 | 30.55 | 41.25 | 45.43 | 9.92 |
| NUEVO LEÓN | 10.25 | 11.60 | 1.59 | 1.37 | 0.88 | 14.70 | 2.20 | 3.02 | 7.23 | 0.99 | 0.63 | 11.76 | 49.22 | 40.92 | 5.00 |
| OAXACA | 15.77 | 12.66 | 3.63 | 5.39 | 1.14 | 16.18 | 3.42 | 3.01 | 25.52 | 1.97 | 2.70 | 33.30 | 43.05 | 44.29 | 9.23 |
| PUEBLA | 14.85 | 16.63 | 2.95 | 3.28 | 1.00 | 16.41 | 2.17 | 3.50 | 23.64 | 2.50 | 3.00 | 31.54 | 40.71 | 48.50 | 8.34 |
| QUERÉTARO | 11.74 | 14.43 | 3.42 | 2.69 | 0.98 | 17.36 | 4.89 | 3.18 | 17.60 | 1.71 | 2.20 | 28.36 | 47.68 | 38.63 | 6.85 |
| QUINTANA ROO | 11.43 | 15.24 | 1.27 | 0.63 | 1.90 | 9.21 | 2.86 | 2.22 | 23.49 | 6.03 | 4.76 | 38.41 | 51.11 | 30.48 | 4.13 |
| SAN LUIS POTOSÍ | 12.21 | 11.47 | 2.72 | 4.79 | 1.65 | 16.01 | 2.81 | 3.55 | 16.09 | 2.06 | 1.82 | 24.34 | 48.10 | 35.07 | 9.90 |
| SINALOA | 12.86 | 19.91 | 2.22 | 3.55 | 0.64 | 21.88 | 2.66 | 3.40 | 19.96 | 2.37 | 1.68 | 32.13 | 40.07 | 48.05 | 9.46 |
| SONORA | 13.73 | 18.84 | 2.04 | 2.84 | 2.27 | 23.16 | 2.72 | 5.68 | 20.32 | 1.48 | 1.02 | 29.51 | 48.35 | 41.32 | 7.15 |
| TABASCO | 15.42 | 21.69 | 1.20 | 2.09 | 1.53 | 20.32 | 1.12 | 2.97 | 13.98 | 3.61 | 1.69 | 21.37 | 42.57 | 50.36 | 5.22 |
| TAMAULIPAS | 13.88 | 15.11 | 2.38 | 1.89 | 1.89 | 18.06 | 2.55 | 3.53 | 9.28 | 1.64 | 0.90 | 16.26 | 49.01 | 42.53 | 5.01 |
| TLAXCALA | 14.21 | 10.70 | 2.46 | 2.81 | 0.88 | 15.09 | 3.68 | 2.63 | 18.95 | 4.21 | 1.93 | 22.98 | 43.33 | 45.79 | 7.54 |
| VERACRUZ | 13.58 | 15.73 | 2.97 | 3.25 | 1.39 | 18.75 | 3.40 | 2.97 | 18.89 | 2.73 | 1.43 | 31.71 | 46.39 | 42.04 | 8.08 |
| YUCATÁN | 10.73 | 16.72 | 1.65 | 2.79 | 1.34 | 18.89 | 1.24 | 2.68 | 13.83 | 3.10 | 2.17 | 22.91 | 47.68 | 40.45 | 7.64 |
| ZACATECAS | 14.39 | 16.21 | 4.01 | 5.10 | 1.82 | 25.14 | 4.74 | 4.92 | 23.13 | 2.19 | 1.46 | 27.69 | 41.71 | 41.89 | 14.03 |

* Dirección General de Epidemiología, Secretaría de Salud (General Direction of Epidemiology, Health Ministry).

**Instituto Nacional de Salud Pública, Encuesta Nacional de la Dinámica Demográfica (National Public Health Institute, National Survey of Demographic Dynamic).

*** Estimations obtained by CONEVAL (Consejo Nacional de Evaluación de la Política de Desarrollo Social, National Council for the Evaluation of Social Development Policy) based on the ENIGH (National Survey of Household Income and Expenditure) 2008, 2010, 2012, 2014, 2016, and 2018.

+ Instituto Nacional de Estadística y Geografía (National Institute of Statistics and Geography).

++ Consejo Nacional del Agua (CONAGUA) (National Water Comission).

+++ Instituto Nacional de Salud Pública (INSP) (The National Institute of Public Health of Mexico).

Communication and Transport) [16]. Finally, information concerning average temperature (˚C) in March 2020 was obtained from the National Council of Water (CONAGUA).

The risk of mortality in tested individuals (MRt) was chosen as the dependent variable based on relevant indicators of COVID-19 epidemiology in Mexico. The number of tests is associated with detection rates. Given the likely under-detection of mild SARS-CoV-2 cases, particularly at the beginning of the outbreak, standardizing deaths by tested cases considers the extent of detection, which could similarly be influenced by structural factors [17]. The remaining variables obtained and calculated from the different data sources were treated as explanatory, except for hospitalization, ICU, and intubation, which we considered as control variables, being an approximate measure of the presence of severe COVID-19 cases and possibly access to services attending COVID-19 in a region.

## COVID-19 MRt estimation by state

To determine the mortality distribution throughout the country, we obtained quantile maps associated with raw and smoothed MRts of COVID-19 cases, and, to consider the different age and sex structures in each state the risks were adjusted for these variables using the average of the 32 states population age and sex compositions as the standard [18]. The risks by state adjusted for sex and age were smoothed using an empirical Bayes estimator, a biased estimator that improves variance instability proper of risks estimated in small-sized spatial units [19]. However, the analyses were performed with both raw and smoothed risks to compare results. We also obtained maps concerning the standardized mortality ratio (SMR) adjusted for sex and age, understanding an SMR, as a comparison of the observed number of events by state to a national standard, the latter using the expected number of events considering as if risks in a state were the same as those at a national level. In all standardization processes, the age structure corresponded to the age groups: 0–9,10–39, 40–69, and 70+, which were derived from the process described in the model selection step.

## Spatial weight estimation and spatial autocorrelation

To determine how much the MRt is spatially associated, first, we obtained queen contiguity weights [20], which consider as neighbours those states sharing at least a point in common, obtaining a squared matrix of dimension 32 with all entries equal to zero or one, where one indicates that two states are neighbours. From these neighbours, weights are calculated by integrating a matrix in a row-standardized form. Moran's I statistic [21] was obtained as a measure of global spatial autocorrelation, and its significance was assessed through a random permutation inference technique based on simulations. To determine regions with similar mortality behaviours local indicators of spatial autocorrelation (LISA) adjusted for sex and age were obtained [22] and used to derive significant spatial clustering through four cluster types: High-High, Low-Low, High-Low, and Low-High. For instance, the Low-Low cluster indicates states with low values of a variable that are significantly surrounded by regions with similarly low values.

## Spatial multivariable linear model

To determine which variables are associated with the MRt and the direction of the association, considering at the same time the spatial nature of the data, a multivariable Generalized Geographically Weighted Regression (GGWR) was fitted [23]. In this model, a dependent variable is measured for each spatial unit. Independent variables (inputs) are simultaneously considered as well, such that the corresponding parameters depend on the coordinates in which the state is spatially located (centroids in the case of polygons); therefore, a parameter is associated

with each state and independent variable. To estimate such a model a weighting diagonal matrix is considered; we used Gaussian spatial weighting to generate it. These weights determine the relationship from any state to another in terms of the distance (Euclidean) between the centroids of states and a bandwidth. The bandwidth determines which spatial units are similar under the GGWR and can be selected using automatized methods; we used a cross-validation (CV) method with an adaptive scheme, i.e. a different bandwidth was used for each unit. Since the response variable is a count (number of deaths), a GGWR with a Poisson distribution and logarithm link function was used, including as offset term the number of people tested with COVID-19 in a logarithmic scale, thus modelling the MRt instead of just the number of deaths.

A global multivariable model for all states, a Poisson multivariable linear model (generalized linear model or GLM) with offset and a logarithmic link function, was also fitted, and significant variables were identified [24]. To obtain the best possible model, including variables with the greatest effect on the risks and satisfying as much as possible all statistical assumptions, we used the following selection process: 1) We fitted univariable Poisson models with offset and logarithmic link function, identified significant effects, and ordered them in absolute value from highest to lowest; 2) We identified variables with good and acceptable linear association with the log-transformed mortality risk by obtaining scatter plots between variables and the transformed risk, including a smoothed LOESS (locally estimated scatterplot smoothing) curve; and, when possible, variables were transformed to improve this assumption, as for GDP as explained above; 3) VIF was used to assess multicollinearity; thus, we fitted a model including variables with acceptable and good linear association, and eliminated any variable with a VIF>10; 4) We added to the resulting model one by one significant variables in the univariable models. First, we added the variable with the highest effect and fitted the associated model identifying whether VIF>10. If not, we added the variable, and if VIF>10, the model was not modified. We proceeded with the resulting model repeating the same process with the second-highest effect; and so on; 5) From this process, we ended with a model consisting of all variables with the greatest univariable effects (0.08 and above in absolute value), better linear behaviour, and without multicollinearity (VIF< = 10); 6) Associations between inputs were observed, mainly, between some explanatory variables and the confounders, but we wanted to obtain the pure effect associated with each explanatory variable after controlling for the confounders. Hence, we used residuals associated with three Poisson models in which ICU, intubated, and hospitalization were used as outputs, including a subset of the variables contained in the fatality risk model chosen in 5) as inputs. This process allowed us to obtain the effects each confounder has over the mortality risk eliminating the effects that explanatory variables have over them. These were confounders as in econometrics or experimental design analyses [25]. Thus, the interpretation of the estimated parameters is not of interest, but they are included to avoid spurious effects associated with the other variables we are interested in once controlling for the severity of the disease, 7) We evaluated goodness-of-fit and validated all model assumptions. For age, we obtained age groups: 0–1, 2–9, 10–19, . . ., 80–89, and 90 years old and over, fitted the corresponding univariable Poisson models with offset, and identified significant age groups and the direction of the association to obtain a new set of age groups: 10–39, 40–69, and 70 and more; and proceeded with these variables as with the others. Moran's I and its significance were obtained for each explanatory variable and confounder in the model to determine if each variable's values were spatially correlated.

The GGWR with the same variables as in the GLM was fitted, and multiplicative effects over the MRts (i.e. the exponentiated estimated parameters) associated with each variable were calculated. Maps associated with these effects were obtained, presenting only those associated with the explanatory variables significant in the global model. To compare the GGWR and

GLM models, the squared sum of residuals (SSR), the coefficient of determination $R^2$, and the Akaike Information Criterion (AIC) were obtained. All statistical analyses were conducted using R version 3.6.2 through the *spdep*, *rgdal*, and *spgwr* packages for spatial analyses and *car* package for the correlation analysis and GeoDa 1.14.0 were also used for some spatial analyses. The significance level for all analyses was 5% (i.e., alpha = 0.05).

## Results

### MRt description and spatial autocorrelation of COVID-19 MRt between states

Maps for quartiles corresponding to the raw and smoothed MRts are similar, except for six states, Durango, Nayarit, and Zacatecas, which from a category using the raw risks move to the next higher quartile in the smoothed risks, the opposite occurring for Coahuila, Mexico, and Veracruz. Only the map concerning the smoothed values is shown (Fig 1A). We observed the largest MRts (raw and smoothed risks above 3.5% and 2.5%, respectively) and the greatest SMR (2.00–4.00) in Baja California, Chihuahua, Quintana Roo, and Tabasco (Fig 1B). Globally, there is a non-significant spatial autocorrelation (Moran's I = -0.054, p = 0.390); however, there is a noticeable Low-Low cluster in the Northeast around San Luis Potosi, two Low-High clusters around Yucatan and Sonora, and one High-Low cluster around Colima (Fig 2). These Low-High clusters make sense since the associated states, especially Sonora, are surrounded by the states with the highest MRts in all the country. We verified that the spatial autocorrelation value and clustering were precisely the same using both the raw values or those obtained with the Bayes spatial technique.

### Fit of multivariable generalized global and linear geographical models for mortality risk from COVID-19 among tested individuals

We found the presence of serious multicollinearity problems through a preliminary analysis obtained by fitting a Poisson multivariable linear model with offset and including all variables since we obtained Variance Inflation Factors (VIF) with values above 50 for some variables. Additionally, a correlation analysis between all input variables was performed (Fig 3A), identifying very correlated variables. Thus, we followed the model selection process, as explained above. Our final global model included as inputs: diabetes (%), obesity (%), GDP, internal and

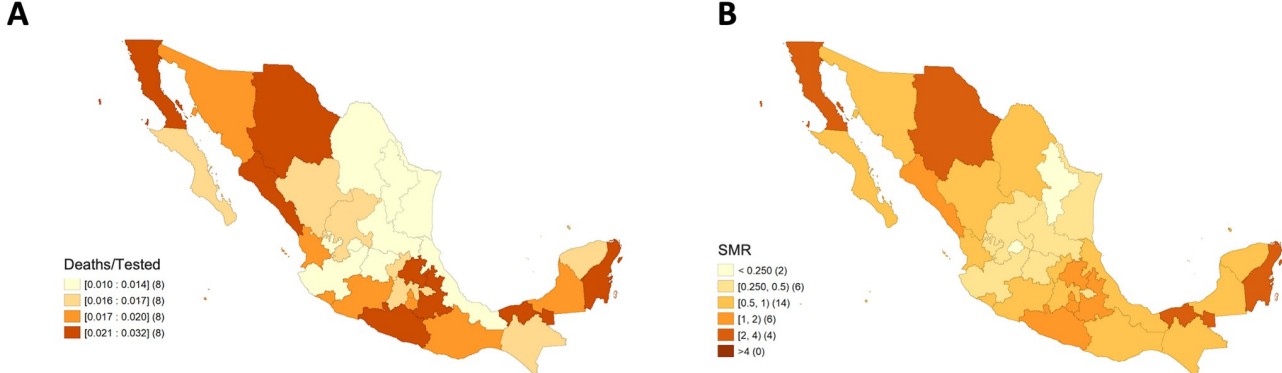

**Fig 1. Maps associated with COVID-19 deaths in Mexico by state until April 21st, 2020, adjusted for sex and age.** A) Quartiles corresponding to mortality risks among tested individuals smoothed through an empirical Bayes procedure. B) Standardized mortality ratio. Shapefile from http://tapiquen-sig.jimdo.com under a CC BY license, with permission from Carlos Efraín Porto Tapiquén, original copyright 2015. Note: The quantities in brackets beside the categories correspond to the number of states in each category.

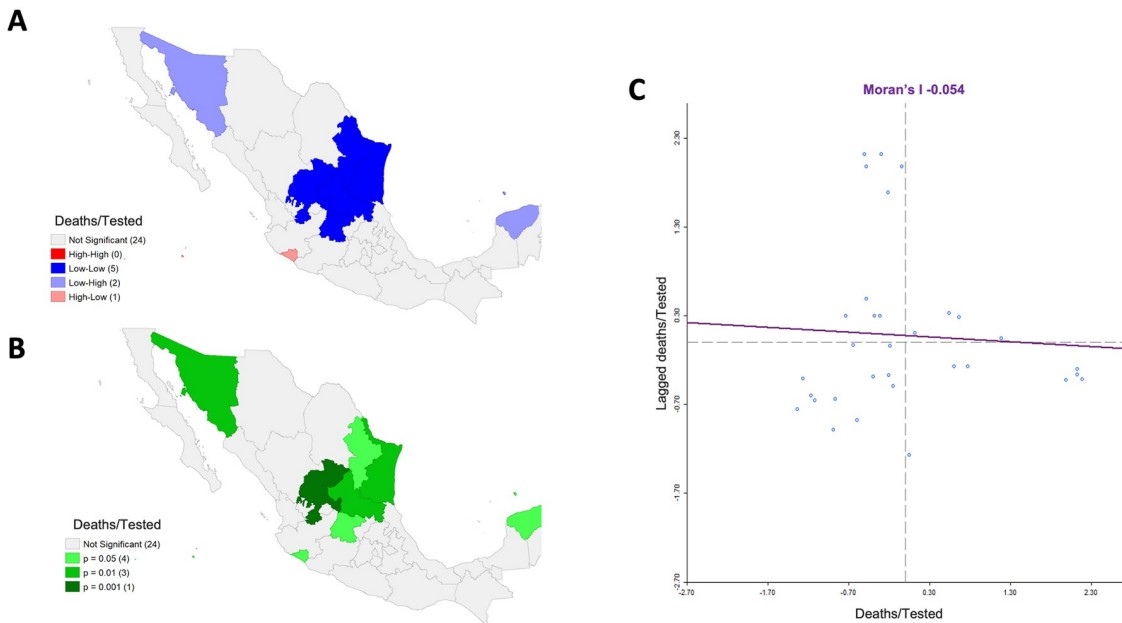

**Fig 2. Spatial clustering associated with mortality risk from COVID-19 among tested individuals adjusted for sex and age in Mexico by state until April 21st, 2020, considering queen contiguity.** A) Significant spatial clustering obtained through Local Indicators of Spatial Autocorrelation (LISA) comparisons. There are four types of clusters: High-High, Low-Low, High-Low, and Low-High, e.g. a Low-Low cluster (blue) indicates states with low values of a variable significantly surrounded by regions with similarly low values. B) P-values associated with the spatial clustering in A), C) Scatter plot associated with the smoothed risks vs their corresponding spatially lagged values, including the associated linear regression fitting, whose slope is the Moran's I statistic. Shapefile from http://tapiquen-sig.jimdo.com under a CC BY license, with permission from Carlos Efraín Porto Tapiquén, original copyright 2015. Note: The quantities in brackets beside the categories correspond to the number of states in each category.

external migration (%), age group of 10 to 39 years (%), physicians-to-population ratio, cardiovascular disease (%), ICU (%), hospitalization (%), and intubated (%). The latter three variables are confounders and, as discussed in the Methods section, to eliminate effects of other variables on them, they are used as residuals associated with appropriate Poisson models with

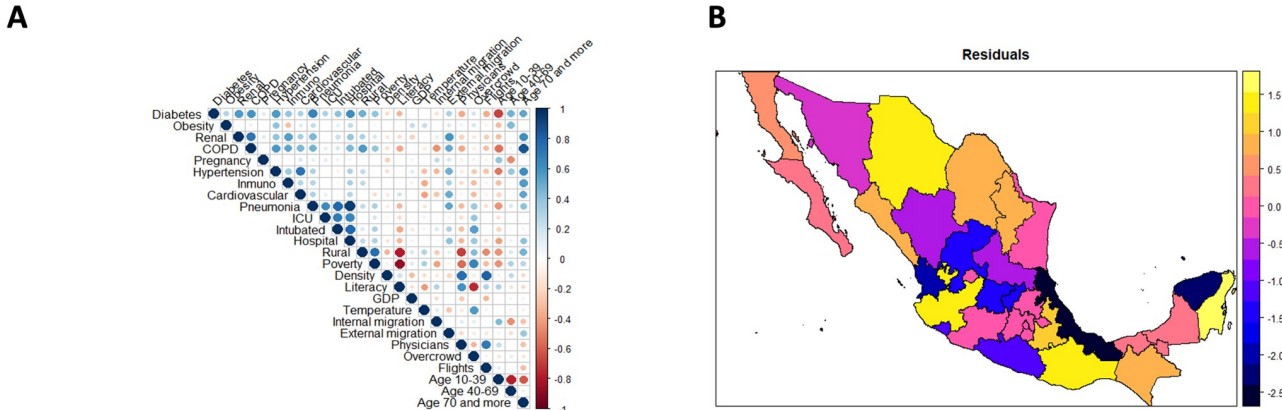

**Fig 3. Figures associated with the selection and goodness-of-fit of a multivariable generalized geographically weighted model (GGWR), with a poison distribution, offset, and a logarithm link function explaining the mortality risk from COVID-19 among tested individuals.** A) Correlation plot including the raw risks and B) Representation of Pearson residuals by state. Shapefile from http://tapiquen-sig.jimdo.com under a CC BY license, with permission from Carlos Efraín Porto Tapiquén, original copyright 2015. Notes: (1) Offset: Log of the total number of people tested in a state; (2) Multiplicative changes in mortality risks (MRt) from the effects of different risk factors by state.

independent variables: diabetes (%), obesity (%), age (%), physicians-to-population ratio, cardiovascular disease (%), plus ICU (%) for the model associated with the intubated variable. Inputs with a significant spatial autocorrelation were the percentages associated with external migration (I = 0.214, p-value = 0.027), age group of 10 to 39 years (I = 0.2, p-value = 0.037), intubated (I = 0.245, p-value = 0.014), and cardiovascular disease (I = 0.359, p-value < 0.001). We found that all variables were jointly significant (LR = 472.19, p-value<0.001). Additionally, through a PP-plot associated with the standardized Pearson residuals and associated Anderson-Darling test (A = 0.260, p-value = 0.689) and scatter plots between the fitted values and standardized residuals, and a similar plot using the root of the residuals instead, we determined that the link function and the way the explanatory variables are related with the response seems correct and there is a good fit. However, we found some overdispersion according to the Chi-squared statistic divided by its degrees of freedom of 1.972, but no significant overdispersion according to an analysis fitting a negative binomial model (p-value = 0.5). There were not any significant pairwise interaction effects between explanatory variables: those between diabetes and all health-related variables, physicians-to-population ratio, and age; those between obesity and all health-related variables; those between GDP and all non-health-related variables; those between migration (internal and external) and non-health-related variables; those between age and all variables and similarly for the physicians-to-population ratio.

A multivariable GGWR with a Poisson distribution, adaptive kernel, and the same input variables was also fitted. The $R^2$ was greater in the GGWR (0.871 vs 0.860) than in the GLM, and the SSR and AIC were both lower (3.35 vs 3.50 in the logarithmic scale and 287.15 vs 308.42, respectively). In Table 3, we summarise the multiplicative effects over the risks, exponentiated parameters, i.e. minimum, quartiles, and maximum, associated with each variable for all states in the GGWR and the global values corresponding to the GLM. Pearson residuals associated with the GGWR are shown in Fig 3B, showing that the worst fit corresponded to two states: Veracruz and Yucatan.

**Table 3. Statistics by variable (minimum, maximum, and quartiles) corresponding to the effects by state\* over the mortality risk from COVID-19 among tested individuals (MRt) under the GGWR and similar effects and p-values associated with a global model (all models consider a Poisson distribution, offset term\*\*, and logarithmic link function).**

| Variable | Min | 1st quartile | Median | 3rd quartile | Max | Global | | | |
|---|---|---|---|---|---|---|---|---|---|
| | | | | | | Value | p-value | 95% Interval | |
| | | | | | | | | Lower | Upper |
| Intercept | 0.004 | 0.005 | 0.005 | 0.005 | 0.005 | 0.005 | <0.001 | 0.000 | 0.056 |
| Diabetes | 1.147 | 1.148 | 1.150 | 1.151 | 1.154 | 1.152 | <0.001 | 1.079 | 1.230 |
| Obesity | 1.121 | 1.122 | 1.123 | 1.125 | 1.125 | 1.122 | <0.001 | 1.081 | 1.166 |
| ICU (residual) | 1.219 | 1.220 | 1.220 | 1.222 | 1.225 | 1.223 | <0.001 | 1.163 | 1.285 |
| GDP (modified) | 1.298 | 1.298 | 1.298 | 1.300 | 1.307 | 1.304 | <0.001 | 1.198 | 1.419 |
| Internal migration | 1.250 | 1.252 | 1.262 | 1.268 | 1.276 | 1.267 | <0.001 | 1.183 | 1.358 |
| External migration | 1.322 | 1.323 | 1.325 | 1.332 | 1.361 | 1.349 | 0.018 | 1.053 | 1.728 |
| Age group 10–39 | 0.907 | 0.907 | 0.908 | 0.909 | 0.910 | 0.908 | <0.001 | 0.873 | 0.944 |
| Intubated (residual) | 1.214 | 1.216 | 1.219 | 1.221 | 1.222 | 1.219 | <0.001 | 1.142 | 1.300 |
| Physicians' ratio | 1.845 | 1.853 | 1.856 | 1.859 | 1.865 | 1.866 | <0.001 | 1.625 | 2.143 |
| Cardio vascular | 0.948 | 0.951 | 0.953 | 0.954 | 0.955 | 0.951 | 0.478 | 0.828 | 1.092 |
| Hospitalization (residual) | 0.957 | 0.957 | 0.957 | 0.959 | 0.960 | 0.958 | 0.009 | 0.928 | 0.990 |

\* Multiplicative changes in the risks (MRt) from the effects of the different risk factors by state.

\*\* Offset: Log of the total number of people tested in a state.

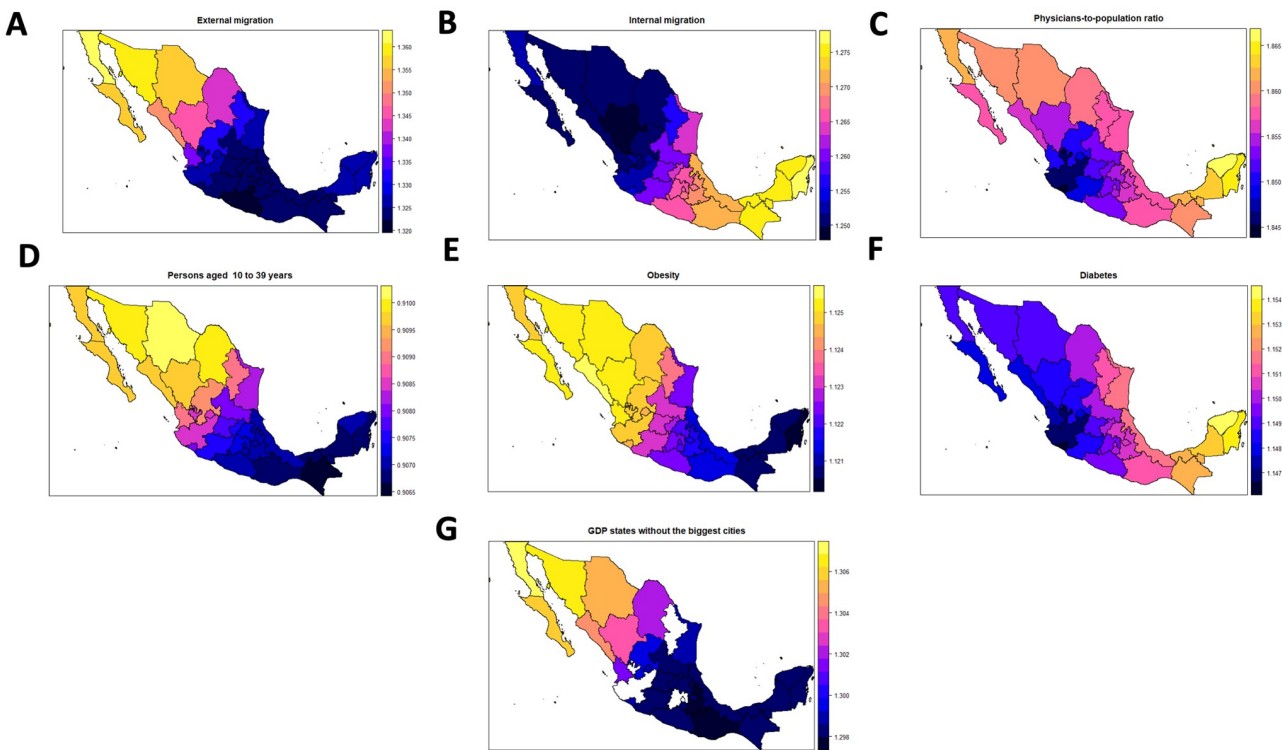

**Fig 4. Multiplicative estimated effects over the tested mortality risk due to COVID-19 among tested individuals (MRt) under a GGWR for those variables that are significantly associated with the response under a global model (Poisson models with offset and a logarithmic link function).** Shapefile from http://tapiquen-sig.jimdo.com under a CC BY license, with permission from Carlos Efraín Porto Tapiquén, original copyright 2015. Note: (1) Offset: Log of the total number of people tested in a state; (2) Multiplicative changes in mortality risks (MRt) from the effects of different risk factors by state.

### Predictors of COVID-19 spatial mortality in Mexico

The exponentiated estimated parameters under the GGWR for each state were obtained for each variable (not shown), and maps were obtained based on the.shp file provided in [26]. Nevertheless, we only present the maps for those explanatory variables significantly associated with the log-transformed MRt in the global model (GLM) (Fig 4). These significant variables were diabetes (%), obesity (%), GDP, internal and external migration (%), age group of 10 to 39 years (%), and physicians-to-population ratio. However, care should be taken when the map for GDP is interpreted considering this variable corresponds to GDP on states not having the biggest cities, as an interaction term between GDP and a binary variable, thus having a fixed value of zero in four states, which are represented as blank spaces in the map. All estimated terms were interpreted by considering fixed values for all variables except the one being interpreted.

### Age and metabolic predictors of MRt in Mexico

Prevalence of obesity has a global (for all states) significant positive association (multiplicative effect) with the COVID-19 MRt (1.122; 95%CI 1.081–1.166). Locally the effect is between 1.120 in Quintana Roo to 1.125 in Sinaloa, having a similar effect on all Mexico, though slightly larger on the North and Centre. For diabetes (%), there is also a significant positive association with the MRt (1.152; 95%CI 1.079–1.230) with local effects between 1.147 in Colima to 1.154 in Yucatan, having a positive effect in all Mexico; but especially in the Centre, South, and

Yucatan peninsula. On the other hand, the proportion of individuals between 10 and 39 years old has a significant negative association with the COVID-19 MRts (0.907; 95%CI 0.873–0.944), locally the effect is between 0.907 in Chiapas to 0.910 in Chihuahua, thus having a similar association in all the country.

## Mobility and socio-economic predictors of MRt in Mexico

The percentage of internal migration in the spatial unit a patient comes from has a significant positive association with the COVID-19 MRts (1.267; 95%CI 1.183–1.356). Locally the effect is between 1.250 in Durango to 1.276 in Quintana Roo, having more effect in the South, centre, and Yucatan peninsula. The percentage of external migration in the spatial unit a patient comes from is also significantly positively associated with the COVID-19 MRt (1.349; 95%CI 1.052–1.728). Locally the effect is between 1.322 in Morelos to 1.361 in Baja California. This effect exists throughout Mexico, but it is more robust on the North and Baja California peninsula. The physicians-to-population ratio is also significantly positively associated with the MRt (1.866; 95%CI 1.625–2.142), with local effects between 1.845 in Colima to 1.865 in Yucatan, having a slightly larger effect on the North and Yucatan peninsula. Finally, GDP excluding the states having the biggest cities in the country (Nuevo León, México, Ciudad de México, and Jalisco) is also significantly positively associated with the MRt (1.304; 95%CI 1.198–1.429), locally the effects are between 1.298 in Oaxaca and 1.307 in Baja California.

## Discussion

In the present study, we demonstrate the relevance of spatial statistical analyses to understand how MRts are distributed throughout Mexico, the presence of spatial clusters related to COVID-19 MRts, the type of associations (negative or positive) with some independent variables and how these associations varied among states. Through our analysis, we were able to identify spatial units or regions in which care could have been considered in terms of mortality due to SARS-CoV-2. Identifying such areas could allow us to understand how these risks were distributed in the early stages of the outbreak.

Considering the global results, variables significantly associated with an increase in the COVID-19 MRt include variables associated with diabetes, obesity, external and internal migration, physicians-to-population ratio, and GDP in states that do not include one of the four largest cities in Mexico. Meanwhile, the proportion of individuals between 10 to 39 years old is significantly associated with a decrease in the MRt of COVID-19, which agrees with previous results analyzed using Mexican data. Cardio-metabolic diseases are associated with adverse COVID-19 outcomes and worst prognosis, probably since they are linked to chronic inflammation, which may synergize with the cytokine storm [27].

Regarding internal and external migration (%), these variables have a solid association with COVID-19 mortality among tested individuals, being external migration the variable with the second-highest positive association. This finding could be related to infectious diseases incidence given that greater movements of people transport pathogens from a geographical region to another, such as for *Mycobacterium tuberculosis* and/or HIV outbreaks [28–30]. Another important variable is the physicians-to-population ratio, which is related to urbanization, population density, and poverty. In this context, from the correlation plot, it can be seen that there is a large positive association between physicians-to-population ratio, population density, and overcrowded households (%), and a large negative association between this variable and both the rural and poverty proportions by state.

On the other hand, the positive association between GDP in states that do not include one of the four largest cities and the MRt might be related to economic activities of these states that

prevent adequate self-isolation measures and, unlike the states with the largest cities, do not have the health infrastructure to prevent death among clinical cases.

The greatest MRts, both raw and smoothed, corresponded to the states of Chihuahua, Quintana Roo, Tabasco, and Baja California (raw values of 4.84%, 4.55%, 4.27%, and 3.96%, respectively, observations until April 21st, 2020). It is noticeable that SMRs for such states are between 2 and 3, suggesting that the risks are above what is nationally expected in these states. We observed a spatial cluster concerning states with low risks; however, at least until the period of our study, there was no significant clustering of states with high risks using the LISA technique.

The fact that Quintana Roo is an important touristic centre might explain the higher number of COVID-19 cases. However, according to our models, the elevated MRt in this state is most strongly associated with the levels of diabetes, internal migration, and the physicians-to-population ratio, whose possible interpretation was discussed above. In Chihuahua and Baja California, the variables with a particularly strong positive association with MRt were external migration (%), obesity (%), and GDP. Meanwhile, in Tabasco, these variables corresponded to diabetes (%) and internal migration (%).

In terms of local effects, the physicians-to-population ratio (heavily associated with urbanicity, overcrowding households, population density, and less poverty) is the variable with the highest positive association with the MRt, but it has a relatively similar strength of association across all of Mexico. External migration (%) has the second-highest association with the MRts, particularly in those states on the north side of the country, in which the risks were the highest. In contrast, internal migration has the fourth-highest association, particularly in the Centre, South, and Yucatan peninsula. The association of the percentages of obesity and diabetes with MRts is similar for both; for obesity, there is relatively a similar association in all the country. Nevertheless, for diabetes, there is a slightly higher association in the Centre, South, and Yucatan peninsula. The associations have also been reported in other studies in other geographical regions [31–33], and although it is still under study, they may be associated with inflammatory stages [34]. Finally, the proportion of individuals between 10 and 39 years old has a relatively similar effect throughout Mexico.

Economic effects, such as poverty, might be better studied in a disaggregated model, including it as a measure at the individual level. Unfortunately, such information is unavailable in the epidemiological data set, and at most, information concerning poverty at a municipality level could be attached to each individual, since the state is the spatial unit containing municipalities. However, we would still be using aggregated values, and in the methodology used to calculate poverty in Mexico, the state values are estimators obtained from a representative sample, whereas the municipality values are estimated through small area estimation techniques; thus, making the former more reliable. In this sense, all analyses were performed at a state instead of at a municipal level. The reason behind this decision is that there were many municipalities with zero values in the early stages of the pandemic, which is the focus of our analysis, and this would have resulted in many issues concerning model fit and convergence considering the probability distributions available for GGWR and other spatial linear models. To obtain similar results, we would have required different tools, such as zero-inflated geographical models, that are not currently available. Future studies could focus on the characteristics of municipalities with zero cases and mortalities, and focus on spatio-temporal analyses of these COVID-19 data.

Our results are robust in terms of the models fitted since most of the statistical assumptions were satisfied; and, though numerically there could be some overdispersion, after fitting a quasi-Poisson and a negative binomial linear model, we obtained similar results and no significant overdispersion according to the latter model. It is essential to notice that we are studying

fatality risks associated with those individuals tested for the disease (MRt); thus, care should be taken if results want to be extrapolated. However, using the projected population in 2020 [35], instead of the tested individuals to model the mortality risks, we obtained similar results. We think that an analysis over the population is somewhat inaccurate because all health-related variables correspond to prevalence in individuals in the data set, which do not necessarily agree with those in the population, the same issue was present for age distribution. If we used population values by state, we would waste all the epidemiological data, except for the number of deaths. Additionally, in the analysis, the number of infected and/or deaths is even more poorly estimated when analyzing the early spread of the disease since, as in many countries, there was a highly selected nature of early tests.

It is essential to mention that a limitation in our results is that the associations should not be extrapolated to lower aggregation levels or individuals due to the potential for ecological bias [36]. Despite these limitations, we were able to identify some spatial predictors of mortality risks associated with COVID-19 at an early stage of the pandemic.

In conclusion, metabolic diseases (%), internal and external migration (%), physicians-to-population ratio, GDP per capita in states without the biggest cities, and the proportion of individuals in the age group between 10 to 39 years old were significantly associated with early COVID-19 mortality risks in Mexico. These predictors likely influence the growth of the pandemic moving forward, but variables, such as the prevalence of metabolic diseases, cannot be easily modified in the short-term. However, identifying variables in Mexico associated with the risks and in specific geographical areas could help in the identification of public policies that could limit the impact of future epidemics. Even though our study focused on the early stages of SARS-CoV-2 spread, its results allow us to understand how the pandemic evolved within Mexico and the possible measures that should be taken to control additional waves of COVID-19 or similar diseases in Mexico and specific zones of the country.

## Author Contributions

**Conceptualization:** Ricardo Ramírez-Aldana.

**Data curation:** Ricardo Ramírez-Aldana, Omar Yaxmehen Bello-Chavolla.

**Formal analysis:** Ricardo Ramírez-Aldana.

**Funding acquisition:** Carmen García-Peña.

**Investigation:** Ricardo Ramírez-Aldana, Juan Carlos Gomez-Verjan.

**Methodology:** Ricardo Ramírez-Aldana.

**Resources:** Ricardo Ramírez-Aldana, Juan Carlos Gomez-Verjan, Carmen García-Peña.

**Software:** Ricardo Ramírez-Aldana, Omar Yaxmehen Bello-Chavolla.

**Supervision:** Ricardo Ramírez-Aldana.

**Validation:** Ricardo Ramírez-Aldana.

**Visualization:** Ricardo Ramírez-Aldana, Omar Yaxmehen Bello-Chavolla.

**Writing – original draft:** Ricardo Ramírez-Aldana, Juan Carlos Gomez-Verjan.

**Writing – review & editing:** Ricardo Ramírez-Aldana, Juan Carlos Gomez-Verjan, Omar Yaxmehen Bello-Chavolla, Carmen García-Peña.

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
