## [Decision Letter · Decision Letter 0]

16 Oct 2020

PONE-D-20-26970

Modelling the impact of health-related variables, migration, and socio-economic factors in the geographical distribution of early COVID-19 mortality in Mexico

PLOS ONE

Dear Dr. Ramírez-Aldana,

Thank you for submitting your manuscript to PLOS ONE. After careful consideration, we feel that it has merit but does not fully meet PLOS ONE’s publication criteria as it currently stands. Therefore, we invite you to submit a revised version of the manuscript that addresses all the issues raised by the reviewers.

We look forward to receiving your revised manuscript.

Kind regards,

Agricola Odoi, BVM, MSc, PhD, FAHA, FACE

Academic Editor

PLOS ONE

Journal Requirements:

2.PLOS journals require authors to make all data necessary to replicate their study’s findings publicly available without restriction at the time of publication. Authors must share the “minimal data set” for their submission. PLOS defines the minimal data set to consist of the data required to replicate all study findings reported in the article, as well as related metadata and methods (https://journals.plos.org/plosone/s/data-availability). Additionally, PLOS requires that authors comply with field-specific standards for preparation, recording, and deposition of data when applicable.

    The values behind the means, standard deviations and other measures reported;

    The values used to build graphs;

    The points extracted from images for analysis.

Authors do not need to submit their entire data set if only a portion of the data was used in the reported study. Also, authors do not need to submit the raw data collected during an investigation if the standard in the field is to share data that have been processed.

As such, we ask you to please ensure that you have met PLOS criteria for the minimal data set.

3.We note that [Figure(s) 1, 2, 3 and 4] in your submission contain [map/satellite] images which may be copyrighted. All PLOS content is published under the Creative Commons Attribution License (CC BY 4.0), which means that the manuscript, images, and Supporting Information files will be freely available online, and any third party is permitted to access, download, copy, distribute, and use these materials in any way, even commercially, with proper attribution. For these reasons, we cannot publish previously copyrighted maps or satellite images created using proprietary data, such as Google software (Google Maps, Street View, and Earth). For more information, see our copyright guidelines: http://journals.plos.org/plosone/s/licenses-and-copyright.

1.    You may seek permission from the original copyright holder of Figure(s) [1, 2, 3 and 4] to publish the content specifically under the CC BY 4.0 license. 

Reviewers' comments:

Reviewer's Responses to Questions

**Comments to the Author**

1. Is the manuscript technically sound, and do the data support the conclusions?

Reviewer #1: Partly

Reviewer #2: Partly

2. Has the statistical analysis been performed appropriately and rigorously? 

Reviewer #1: Yes

Reviewer #2: I Don't Know

3. Have the authors made all data underlying the findings in their manuscript fully available?

Reviewer #1: Yes

Reviewer #2: Yes

4. Is the manuscript presented in an intelligible fashion and written in standard English?

Reviewer #1: Yes

Reviewer #2: Yes

5. Review Comments to the Author

Reviewer #1: This paper examines the factors associated with early COVID-19 mortality in Mexico at the state level. The authors find that both internal and external migration are important, as are health risk factors (counts of diabetes, hypertension, and ICU hospitalization). Pregnancy rates are negatively associated with early COVID-19 mortality.

There is much to like about this paper. The fact that it focuses on a large, highly heterogeneous, upper middle country makes it inherently interesting, especially as Mexican health and social statistics data quality are both high. Even more importantly, the paper is meticulous in correcting for spatial autocorrelation, using generalized geographically-weighted regression techniques and, since they are estimating counts, a Poisson regression approach.

At the same time, I have several problems with the analysis. In rough order of descending importance:

1. (page 5) A key problem with analyzing the early spread of COVID-19 is that prevalence rates were poorly estimated/numbers infected were not well known. This may have extended to deaths as well, and this problem needs careful discussion.

2. (page 5) The authors use ICU hospitalizations as a measure of case load severity. However, ICU numbers also reflect access, so that two very different effects are incorporated into a single term, making it difficult to place any meaning on the estimated coefficient.

3. ICU admissions are also endogenous: in effect, one would be better off estimating them as well in a two-stage process.

4. More generally, the authors include a wide range of variables as potential explanatory variables without building a model and explaining how these variables will affect mortality – whether they will do so directly or indirectly; whether the terms represent a factor that matters directly or whether they serve as proxies for omitted/unmeasurable terms. Given this absence, there can be no causal interpretation of the results.

5. (page 12) Independent variables highly correlated with one another “and less correlated with the response” were removed. A better approach might have been to have estimated principal components and used those.

6. The analysis here is a snapshot of early mortality while correcting for spatial autocorrelation. It might have been more interesting and informative to have explored early transitions using multiple time periods and capturing the interregional spread.

7. (page 6) The authors include measures of poverty and housing density. In fact, these terms are likely to matter far more for urban areas, and hence it would be better to use strictly urban measures of these terms.

8. (page 8, lines 157) It isn’t clear to me whether some of these measures are counts or rates. They should be defined precisely.

9. (page 5) the authors group the population into only two age classes (65+ or under 65). This is a strong assumption and adding additional age delineations in robustness check regressions would make sense.

10. (page 16) In particular, there is no causal meaning in the finding that the percentage of pregnant women is associated with a 22-23% decrease in COVID-19 counts. The idea that pregnancy confers a “protective effect,” while possible at the individual level, seems unlikely to extend to the whole population, and the fraction of the Mexican population that is pregnant is likely too small to have a measurable effect at the aggregate level.

11. (page 12) Although spatial autocorrelation is addressed, linkages due to air travel (both internal and international) may be important and should be considered.

In short, there is a fair amount to be addressed. It is important to do so: there are vast numbers of papers on COVID-19 mortality and transmission coming out, and only a few will be read or cited in a couple years. It is worth taking the care and effort to ensure that the paper falls into this elite group.

There are minor typos that need to be corrected (for example, on line 3 of the abstract, it should read > 65, not <65; on page 4 line 75, note that the population pyramid is flattening rather than widening as Mexico ages). Beyond that, the paper is well-written, except that the table and figure titles are far too long.

Reviewer #2: Manuscript #: PONE-D-20-26970

Title: Modelling the impact of health-related variables, migration, and socio-economic factors in the geographical distribution of early COVID-19 mortality in Mexico

Authors: Ramirez-Aldana et al.

General comments:

The researchers have an interesting dataset and applied a variety of spatial techniques to understand the distribution and state-level risk factors for COVID-19 in Mexico. Their manuscript has the potential to make an important contribution to our understanding of the disease, but I have the following major epidemiological concerns about their analytical approach:

1. Currently, most epidemiologists distinguish between risks (expressed as percentages) and rates (expressed as number of cases per person-years). While the authors state in some areas that the outcome variable is mortality rate, they express this outcome as a risk. There is also an issue with their denominator for measuring mortality risk/rate. The denominator should be the state population if they are measuring the mortality risk or state-level person-years for mortality rate. However, it appears that they are actually using the number of individuals who tested positive for COVID-19 (although this needs to be stated more clearly). As a consequence, they are modeling the case-fatality risk. This is a legitimate measure, but this needs to be clarified throughout the manuscript and issues associated with this epidemiological measure should be addressed. The mortality risk and rate are substantially lower than the case-fatality risk at this time, and the incorrect use of these terms will contribute to confusion in the literature.

2. The researchers appear to have a great deal of high quality patient-level data that they have aggregated for the purpose of fitting a state-level GGWR model. I would suggest that the authors consider fitting a mixed logistic regression model with a random intercept for state where they can examine associations with patient-level and contextual state-level variables. Age and sex are important for predicting case-fatality and the interpretation of these variables is subject to ecological bias when these individual level variables are averaged for what I assume are the COVID-19 positive patients in each state. In addition, based on the Moran’s I result, there does not appear to be strong justification to be concerned about spatial auto-correlation. It is also unclear if the GGWR model is appropriate for area data where spatial distances are measured between centroids of relatively large geographical areas.

3. The authors’ model fitting process does not appear to consider issues like confounding and interaction effects. This should be addressed in their methods and results.

4. The authors may want to consider using Kulldorff’s spatial scan statistic or Tango and Takahashi’s flexible scan statistic (both available in R and a freeware) for high and or low cluster detection especially if they want to consider clusters of varying sizes and/or shapes.

Specific comments:

Please note that based on the above major concerns, I will only provide specific comments that might assist in revising the manuscript.

i. Please edit the title so it is clear that case-fatality risk is being explored or make appropriate changes to the dependent variables in all the analyses.

ii. Line 29. “Torpid” means inactive or lethargic. I don’t believe this is what the authors mean to say.

iii. Please make certain footnotes for all tables and figure legends are in English or include English translations. Please upload figures of higher resolution. They were very difficult to read. Please note on Table 1 that “pneumonia” is misspelled in the column heading and make certain any short forms have appropriate footnotes (e.g., “Age_d”). In addition, footnotes might be needed for several column headings. For example, it is unclear what the "Number of residents" represents. Is it the population in units of 1000, the number of residents infected with the virus, or the number tested?

iv. Please use the term “multivariable” rather than “multivariate” to indicate a model with multiple independent variables. The current recommended convention is to use the term multivariate for statistics that have more than one dependent variable.

v. Make certain in the methods that it is stated how the linearity assumption between the independent variable and the outcome was assessed for any models fitted, and clearly state which diagnostics were performed for all models fitted.

vi. Please clarify on line 189 that “number of people tested with COVID-19” indicates the number of people that tested positive for COVID-19.

vii. In line 203, the authors should state “The significance level for all analyses was 5% (i.e., alpha=0.05)”.

viii. In line 215, the authors state they used a “Bayes spatial technique”. Do the authors mean they performed the clustered analyses on data smoothed using a Bayesian technique or some other Baysian process to estimate the p-values? This should be clarified in the methods, and I would strongly recommend against performing cluster-detection methods on smoothed data since the smoothing process can create clusters.

ix. Figure 1. Please note that relative risk and excess risk are not synonyms. Excess risk is a synonym for risk difference.

x. Line 236. The authors should explain that VIF was initially used to assess multicollinearity in the methods. In the methods, they should state what threshold was subsequently used to determine if two independent variables were highly correlated. I would suggest assessing the univariable statistical significance of each variable as part of the process of considering variables for inclusion in the multivariable model.

xi. Please make sure to state the measure of association estimated from the exponentiated coefficients and present them in table form.

xii. Lines 278-308. Avoid the use of the phrases “x% increase” or “x% decrease” when describing measures of associations (e.g., incident rate ratios, relative risks, or odds ratios). They are ratio measures and the value after subtracting from 1 (e.g., 1.73-1) can vary depending on the referent category being used for categorical variables or the direction of change for continuous variables. For instance, if the odds ratio for sex was 4 with females as the referent, would the authors be comfortable stating that the odds were 300% greater (i.e., 4-1/1=3) for males than females, but if the referent were males, making the OR= 0.25 (i.e., OR=1/4), state the risk was decreased 75% for females (0.25-1/1)? I would recommend simply stating if the odds, risk, or rate was significantly higher or lower for one category vs. another, or stating whether there was a significant positive or negative association for linear continuous variables. The authors can refer to a table for specific values or include the measure of association, its 95% CI, and p-value in brackets beside the statement.

6. PLOS authors have the option to publish the peer review history of their article (what does this mean?). If published, this will include your full peer review and any attached files.

Reviewer #1: No

Reviewer #2: No

---

## [Author Response · Author response to Decision Letter 0]

9 Dec 2020

Reviewer #1: This paper examines the factors associated with early COVID-19 mortality in Mexico at the state level. The authors find that both internal and external migration are important, as are health risk factors (counts of diabetes, hypertension, and ICU hospitalization). Pregnancy rates are negatively associated with early COVID-19 mortality.

There is much to like about this paper. The fact that it focuses on a large, highly heterogeneous, upper middle country makes it inherently interesting, especially as Mexican health and social statistics data quality are both high. Even more importantly, the paper is meticulous in correcting for spatial autocorrelation, using generalized geographically-weighted regression techniques and, since they are estimating counts, a Poisson regression approach.

Response: Thank you very much for the revision. We really appreciate your time in reviewing and commenting our paper. We have modified the paper attending to all your suggestions. A point-by-point response to every raised issue is provided as an additional file included in the submission.

Reviewer #2: Manuscript #: PONE-D-20-26970

Title: Modelling the impact of health-related variables, migration, and socio-economic factors in the geographical distribution of early COVID-19 mortality in Mexico

Authors: Ramirez-Aldana et al.

General comments:

The researchers have an interesting dataset and applied a variety of spatial techniques to understand the distribution and state-level risk factors for COVID-19 in Mexico. Their manuscript has the potential to make an important contribution to our understanding of the disease, but I have the following major epidemiological concerns about their analytical approach.

Response: Thank you very much for the revision and for your time and effort in reviewing our paper. We have modified the paper attending to all your comments and suggestions. A point-by-point response is provided as an additional file included in the submission.

---

## [Decision Letter · Decision Letter 1]

12 Jan 2021

PONE-D-20-26970R1

Modelling the impact of health-related variables, age, migration, and socio-economic factors in the geographical distribution of early tested case-fatality risks associated with COVID-19 in Mexico

PLOS ONE

Dear Dr. Ramírez-Aldana,

Thank you for submitting your manuscript to PLOS ONE. After careful consideration, we feel that it has merit but does not fully meet PLOS ONE’s publication criteria as it currently stands. Therefore, we invite you to submit a revised version of the manuscript that addresses all the issues raised by reviewer 2.

We look forward to receiving your revised manuscript.

Kind regards,

Agricola Odoi, BVM, MSc, PhD, FAHA, FACE

Academic Editor

PLOS ONE

Reviewers' comments:

Reviewer's Responses to Questions

**Comments to the Author**

1. If the authors have adequately addressed your comments raised in a previous round of review and you feel that this manuscript is now acceptable for publication, you may indicate that here to bypass the “Comments to the Author” section, enter your conflict of interest statement in the “Confidential to Editor” section, and submit your "Accept" recommendation.

Reviewer #1: All comments have been addressed

Reviewer #2: (No Response)

2. Is the manuscript technically sound, and do the data support the conclusions?

Reviewer #1: Yes

Reviewer #2: Partly

3. Has the statistical analysis been performed appropriately and rigorously? 

Reviewer #1: Yes

Reviewer #2: Yes

4. Have the authors made all data underlying the findings in their manuscript fully available?

Reviewer #1: Yes

Reviewer #2: Yes

5. Is the manuscript presented in an intelligible fashion and written in standard English?

Reviewer #1: No

Reviewer #2: Yes

6. Review Comments to the Author

Reviewer #1: This paper examines the factors associated with early COVID-19 tested case fatality in Mexico at the state level. The authors find that both internal and external migration are important, as are health risk factors (counts of diabetes, hypertension, and ICU hospitalization). Physician density also has a strong positive association.

The authors have satisfactorily addressed all of the issues that I and the other reviewer raised. I’m impressed, both because together we had a very large number of comments, and because of the excellent, detailed responses. The resulting paper and its focus on t-CFRs strikes me as being a significant improvement. The limitations that remain either will be addressed in parallel or subsequent work (for example, using individual rather than grouped observations) or are inherent and cannot be fully addressed (specifically, the highly selected nature of early tests).

My remaining comments are secondary:

1. Most important of all, the paper needs to be reviewer by a native writer of English. Every page has awkward phrasings, though there are few if any grammatical errors. The last paragraph in the paper (lines 432-478) needs to be broken up into several pieces.

2. Line 72: note that OCDE is abbreviated as OECD in English

3. Line 76: note that population aging makes the pyramid become more rectangular rather than flattened.

4. Line 223: “the corresponding estimated parameters do not have a meaning” sounds jarring and needs to be restated. Indeed, while I am happy with the use of residuals from prior regressions, the rationale for doing so should be explained. This leads to a bigger issue for the authors and the editor: the paper lays out a lot of technique without explaining why various approaches are being taken. This is fine if the audience consists of epidemiologists (or quantitative demographers or econometricians), but if the readership is more general, they are likely to treat the techniques as a black box. Of course, lots of detail adds to length, which is also costly.

5. It is important (around line 400) to be careful about the use of causal language. For prior comorbidities (diabetes, obesity), this makes sense. It is less clear in the case of migration (as migratory links might lead people to travel for health care) but still fairly plausible. It is implausible in the case of variables like physician density of GDP/capita, since diagnosis and testing will be different.

Reviewer #2: Manuscript #: PONE-D-20-26970-R1

Title: Modelling the impact of health-related variables, age, migration, and socio-economic factors in the geographical distribution of early tested case-fatality risks associated with COVID-19 in Mexico

Authors: Ramirez-Aldana et al.

General comments:

The authors have done a very thorough job responding to reviewers’ questions and suggestions. However, the following issues need to be addressed in the revised manuscript:

1. The authors spend part of the discussion writing about variables measured at different hierarchical levels. In their response letter to the first review, they do not seem to recognize that multi-level models not only include random intercepts at various hierarchical levels but also allow for the proper estimation of variables measured at these different levels (e.g., individual person, municipality, and state). One of the major limitations of their modeling technique (geographic weighted regression) for their study is that it required the authors to take individual level variables and make them ecological variables (e.g., age of the individual is now the proportion of the sample in an age category). Consequently, all their results are subject to ecological bias (i.e., the ecological fallacy). Similarly, the ability to detect interaction effects is reduced in ecological studies. However, if the authors revise the discussion to include issues related to ecological bias, I think that would be sufficient. It is unfortunate that the GWR approach was not more revealing. I suspect most if not all of the variation in variable coefficients attributed to differences among states would have fallen within the 95% CI of the estimates from the global model.

The authors may wish to review material from the Centre for Multi-level Modeling (http://www.bristol.ac.uk/cmm/learning/multilevel-models/what-why.html) and other references where multi-level models have been used for epidemiological studies. However, based on their objectives, I do not expect the authors to change their modeling approach.

2. If the authors are committed to presenting their GWR model, they should include the results of the test for bandwidth or other tests to support that the GWR model fits better than the “global model”. In addition, the authors should present the tests of non-stationarity for each variable. Only if there’s evidence of non-stationarity should the authors present results suggesting that there are different effects among states for a particular variable.

3. In terms of their risk maps, the authors should strongly consider adjusting for sex and age (e.g., indirect standardization). Currently, the maps may only be showing demographic differences among the people tested within these states. Similarly, the LISA statistics should adjust for age and sex. I believe this can be done with LISA and it certainly can be done with the spatial scan statistic.

4. In terms of terminology, technically an observed/expected ratio is referred to as a standardized morbidity/mortality ratio (SMR) rather than a relative risk. Please note that using indirect standardization, an SMR can be estimated that accounts for age and sex distribution in estimating the expected count.

5. Based on the authors’ response, it is now clear they are modelling the risk of mortality from COVID -19 among all tested individuals not just those that tested positive for COVID-19. Consequently, it would be clearer if they used the phrase mortality risk from COVID-19 among tested individuals and use “MRt” as an acronym and avoid the current “t-CFR” since a case-fatality risk refers to the risk of mortality among those who have the disease not those tested for the disease. Even with the definition provided in the current text, altering the meaning of a commonly used epidemiological term will confuse many readers.

6. In the discussion, the authors should avoid commenting on aspects of COVID-19 that are not examined in their study. For instance, risk factors for death may not be the same as those for disease so make certain associations are explained in this context. Similarly, avoid comments about the propagation of disease (e.g., lines 357-359) when space-time clusters were not examined nor were space-time factors considered in the GWR models.

7. The authors have included descriptions in the discussion of a large number of analyses that are not presented in the methods or results. While these comments were appropriate for their response letter, they should not be included in the current manuscript unless the authors wish to include a comparative methodology component to the manuscript. However, if they choose to add this element, I would encourage them to split the manuscript in two and focus on spatial mapping and cluster detection in one manuscript and the risk factor analyses in another. In each paper, they could include all the methods being considered in the methods section and provide the accompanying results.

8. There is some confusion about the nature of hospitalization, intubation, and ICU use in the authors’ analyses. At the individual-level, these are intervening variables and not confounding variables. Typically, intervening variables are not included in a model and confounders are usually controlled for by their inclusion in the model. The authors might want to clearly provide a definition of a confounding variable and clarify if they are using the epidemiological definition.

9. The authors should be cautious throughout the abstract, results, and discussion about the use of the term “impact”. Measures of effect (e.g., attributable fraction, population-attributable fraction) are often used to measure the impact of controlling for a variable; a variable may have a large relative risk, but its impact on the population may be quite small since the exposure is rare. I would recommend simply commenting on the direction and statistical significance of the measures of association in the text and not ranking the “impact of these variables”.

10. Clarify if you are using a Poisson model to estimate incident rate ratios or relative risks in the methods and in the results.

Specific comments:

Title:

i. Perhaps a better title would be “Spatial epidemiological study of the distribution, clustering, and risk factors associated with COVID-19 mortality in Mexico”.

Introduction:

i. Line 58. I do not believe “Spanish Influenza” needs to be in italics.

ii. Line 72. It should read “according to the OCDE” and “OCDE” should be written in full the first time the acronym is introduced.

iii. Line 77. It would be better to state, “are likely to lead to higher rates of mortality”.

iv. Line 92. Replace “variate” with “vary”.

v. Lines 100-101. It should read, “…in Mexico that should be examined in more detail to better understand…”.

vi. Line 105. Replace “A first set of variables” with “Individual level variables” and then edit the rest of the sentence to reflect this change.

vii. Line 109-110. It should read “who were suspected of having COVID-19”. In a following sentence or within this sentence clarify if testing required suspicion of disease by a physician and/or if members could receive a test based on their own concerns.

viii. Lines 115-116. Make it clear here and throughout the manuscript that you are looking at the proportion of individuals in these age groups.

ix. Line 121. Please replace “rate” with “risk”.

x. Line 126. Please replace “clustering” with “population density in various contexts” to avoid confusion with clustering in the statistical sense.

xi. Lines 133-134. You should avoid dropping data to achieve linearity. If necessary, model the polynomial or categorize the variable (i.e., classify the state GDP into quantiles).

Results:

i. Lines 119-123. The text is very unclear for this section. Please see my comments (general comments section) concerning these variables being intervening variables and edit accordingly or justify the approach used to control for the effect of these variables.

ii. Lines 281-289. Please note that for Poisson models there is no assumption of homogeneity of variance or normality. However, Anscombe residuals will have a normal distribution if the model fits the data. In addition, the null-hypothesis for the overall model p-value is that all coefficients equal zero and is not intended to test for lack of model fit.

iii. Throughout the figures and tables concerning your Poisson models, include a footnote that the offset is the log of the total number of people tested in the state and indicate what measure of association is being estimated.

iv. Table 2. Include the 95% CI for the global effect for each variable.

v. Lines 313-314. Make certain it is clear that variables measured at the individual level have been collapsed by writing “diabetes (%)” rather than just “diabetes”, for example.

vi. Line 350. Are there any diagnostics for GWR models? If so, include them here and describe them in the methods.

Discussion: Please edit based on the suggestions in my general comments.

References: Please make sure article titles are formatted consistently.

Figures: Make sure that the same symbols are used for ranges in categories for all figures and include a footnote that the brackets beside these categories concern the number of states in each category.

7. PLOS authors have the option to publish the peer review history of their article (what does this mean?). If published, this will include your full peer review and any attached files.

Reviewer #1: No

Reviewer #2: No

---

## [Author Response · Author response to Decision Letter 1]

24 Feb 2021

PONE-D-20-26970R1

Modelling the impact of health-related variables, age, migration, and socio-economic factors in the geographical distribution of early tested case-fatality risks associated with COVID-19 in Mexico

PLOS ONE

Dear Dr. Ramírez-Aldana,

Thank you for submitting your manuscript to PLOS ONE. After careful consideration, we feel that it has merit but does not fully meet PLOS ONE’s publication criteria as it currently stands. Therefore, we invite you to submit a revised version of the manuscript that addresses all the issues raised by reviewer 2.

We look forward to receiving your revised manuscript.

Kind regards,

Agricola Odoi, BVM, MSc, PhD, FAHA, FACE

Academic Editor

PLOS ONE

Response: Thank you very much for the revision and for considering our manuscript. We have modi\ffied the paper attending to the comments and suggestions raised by the two reviewers. A point-by-point response is provided as an additional document. The modifi\fcations have been marked in yellow in the new version of the paper.

Reviewer #1: This paper examines the factors associated with early COVID-19 tested case fatality in Mexico at the state level. The authors find that both internal and external migration are important, as are health risk factors (counts of diabetes, hypertension, and ICU hospitalization). Physician density also has a strong positive association.

The authors have satisfactorily addressed all of the issues that I and the other reviewer raised. I’m impressed, both because together we had a very large number of comments, and because of the excellent, detailed responses. The resulting paper and its focus on t-CFRs strikes me as being a significant improvement. The limitations that remain either will be addressed in parallel or subsequent work (for example, using individual rather than grouped observations) or are inherent and cannot be fully addressed (specifically, the highly selected nature of early tests).

Response Thank you very much for your revision and comments. We appreciate your time in reviewing and commenting our paper. We have modified the paper attending to all your suggestions. A point-by-point response to every raised issue is provided as an additional file.

Reviewer #2: Manuscript #: PONE-D-20-26970-R1

Title: Modelling the impact of health-related variables, age, migration, and socio-economic factors in the geographical distribution of early tested case-fatality risks associated with COVID-19 in Mexico

Authors: Ramirez-Aldana et al.

General comments:

The authors have done a very thorough job responding to reviewers’ questions and suggestions.

Response: Thank you very much for your time and effort in reviewing our paper. We have modified the paper attending to all reviewer’s comments and suggestions. A point-by-point response is provided as an additional file.

---

## [Decision Letter · Decision Letter 2]

6 Apr 2021

PONE-D-20-26970R2

Spatial epidemiological study of the distribution, clustering, and risk factors associated with early COVID-19 mortality in Mexico

PLOS ONE

Dear Dr. Ramírez-Aldana,

Thank you for submitting your manuscript to PLOS ONE. After careful consideration, we feel that it has merit but does not fully meet PLOS ONE’s publication criteria as it currently stands. Therefore, we invite you to submit a revised version of the manuscript that addresses all the issues raised by the reviewer.

We look forward to receiving your revised manuscript.

Kind regards,

Agricola Odoi, BVM, MSc, PhD, FAHA, FACE

Academic Editor

PLOS ONE

Journal Requirements:

Reviewers' comments:

Reviewer's Responses to Questions

**Comments to the Author**

1. If the authors have adequately addressed your comments raised in a previous round of review and you feel that this manuscript is now acceptable for publication, you may indicate that here to bypass the “Comments to the Author” section, enter your conflict of interest statement in the “Confidential to Editor” section, and submit your "Accept" recommendation.

Reviewer #2: (No Response)

2. Is the manuscript technically sound, and do the data support the conclusions?

Reviewer #2: Yes

3. Has the statistical analysis been performed appropriately and rigorously? 

Reviewer #2: Yes

4. Have the authors made all data underlying the findings in their manuscript fully available?

Reviewer #2: Yes

5. Is the manuscript presented in an intelligible fashion and written in standard English?

Reviewer #2: Yes

6. Review Comments to the Author

Reviewer #2: Manuscript #: PONE-D-20-26970R2

Manuscript title: Spatial epidemiological study of the distribution, clustering, and risk factors associated with early COVID-19 mortality in Mexico

Authors: Ramirez-Aldana et al.

General comments:

The authors have done a thorough and thoughtful job in addressing reviewers’ questions and concerns in the revised manuscript and response letter. Below I have included editorial suggestions to make the text clearer and have some requests for clarification. In the future the authors need to have their work properly edited.

Specific comments:

1. Line 33: It should read, “Among health related variables,……”

2. Lines 35-36: It should read, “external and internal migration had positive associations……..mortality risk with external migration having the second-highest positive association. As an indirect measure of urbanicity, population density…………………………overcrowding in households……”

Note: “Urbanicity” means the degree to which a given geographical area is urban while “urbanity” also refers to being refined.

3. Line 40: It should read, “10 to 39 years had a negative association with MRt.”

4. Line 44: It should read, “fatality risk varied by zone.”

5. Line 60: It should read, “in nursing homes.”

6. Line 71: Please remove “On the other hand”. It does not properly link the two sentences.

7. Line 73: It should read, “conditions that are now linked to…”

8. Line 84: It should read, “increasing….”

9. Lines 90-92: It should read, “are distributed throughout a territory, detect spatial clusters, and measure how the effects of variables associated with this risk vary within any given territory.”

10. Line 92: It should read, “geo-epidemiological studies….”

11. Lines 93-94: It should read, “infectious diseases, and other conditions,……….health policies and knowledge of disease spread.”

12. Line 95: Replace “derive” with “measure”.

13. Line 99: Replace “for such a task” with “to accomplish these objectives”.

14. Line 101: Replace “risks” with “risk”.

15. Line 104: Replace “considering” with “concerning” and “in Mexico” with “of Mexico”.

16. Line 116: It should read, “used three age groups for subsequent analyses:”

17. Line 130: Replace “are” with “were”.

18. Line 131: Replace “by” with “for”.

19. Line 147: Replace “dying” with “mortality” when you first define “MRt”.

20. Lines 152: Replace “are” with “were”.

21. Table 1: Please make certain the number of decimal places presented is consistent with the precision the variable was measured. Technically, the natural log of the number of tested individuals will be your offset in subsequent Poisson regression models; it may be better to comment on this being used for the offset in a footnote rather than the column heading.

22. Line 168: Replace “along the territory” with “throughout the country” or “among the states of Mexico”.

23. Lines 184-185: Replace “a one” with “one”.

24. Line 204: Replace “states” with “the centroids of states”.

25. Lines 215-240: Replace each “,” between numbered phrases with a “;”. This type of punctuation will make it easier for readers to follow the long number of complex phrases.

26. Line 232: Do the authors mean “after controlling for the confounders”? Please edit accordingly if this is correct.

27. Line 233: It should read, “the other three models…”

28. Line 261: Replace “superior one” with “higher quartile”.

29. Line 265: There is an inconsistency between your text where Moran’s I = -0.059 and your figure where Moran’s I = -0.054.

30. Line 295: Why include the outcome in the matrix of correlations? Multicollinearity is only an issue for the independent variables. I would remove the dependent variable from the matrix unless the authors can provide an explanation for this decision to avoid confusing readers.

31. Lines 307-312: I would suggest the authors review Methods in Epidemiologic Research by Dohoo et al. concerning assessing the fit of Poisson regression models (https://projects.upei.ca/mer/). They may want to fit a negative binomial regression model for their global model to assess if there is really an over-dispersion issue (i.e., assess if alpha is significantly different from zero).

32. Lines 313-314: It should be clear in the methods which interactions were tested.

33. Lines 321-322: Based on the legend, the largest residuals should be in yellow and black, but these do not appear to be the colours of Veracruz and Yucatan in Figure 3b. Please re-check that these regions have the largest residuals.

34. Table 2: Please clarify if these are 95% confidence intervals in the column heading. Usually the terms “upper” and “lower” confidence intervals are used rather than “inferior” and “superior”.

35. Line 373: Replace “overall” with “throughout”.

36. Line 376: Replace “on” with “in” and be consistent with capitalizing the word “north” (see line 373).

37. Line 383: Replace “along” with “throughout”.

38. Lines 384-385: It might be clearer and more succinct to write, “and how the associations with some independent variables varied among states.”

39. Line 388. It might be better to write, “in the early stages of the outbreak.”

40. Lines 389-390: It should read, “an increase in…….include variables…..”

41. Line 392: Replace “greatest” with “largest”.

42. Lines 407-408: It might be better to write, “related to the economic activities of these states that prevent adequate self-isolation measures and, unlike the states with the largest cities, do not have the health infrastructure to prevent deaths among clinical cases.”

43. Line 389: Replace “on” with “in”.

44. Line 390: Replace “percentages” with “variables”.

45. Line 392: Replace “greatest” with “largest”.

46. Lines 415-416: It should read, “at least until the period of our study, there was no significant clustering of states…….”

47. Line 418: Replace “suggests” with “might explain the higher…..”

48. Line 419: Replace “on such” with “in this”.

49. Line 420: It should read, “most strongly associated with the levels of diabetes, internal migration, and the physician-to-population ratio.”

50. Line 422: Do the authors mean, “a particularly strong positive association…”?

51. Line 424: Replace “correspond” with “corresponded”.

52. Line 425: Replace “urbanity” with “urbanicity”.

53. Line 426: Drop “which”.

54. Line 427: It should read, “but it has a relatively similar strength of association across all of Mexico.”

55. Line 433: Replace “little” with “slightly”.

56. Line 434: It should read, “The associations have also….”

57. Line 437: Replace “on” with “throughout”.

58. Line 439: Replace “an” with “the”.

59. Line 441: Replace “, being” with “since the state is”.

60. Line 445: Replace “being” with “making”.

61. Lines 447-455: It should read, “in the early stages of the pandemic, which is the focus of our analysis, and this would have resulted in many issues concerning model fit and convergence. To obtain similar results, we would have required different tools, such as zero-inflated geographical models, that are not currently available. Future studies could focus on the characteristics of municipalities with zero cases and mortalities, and focus on the spatio-temporal analysis of these COVID-19 data.”

62. Line 456: It should read, “models fitted since most of the statistical…..”

63. Line 457: Replace “is” with “was”.

64. Line 464: It might sound better to state, “the same issue was present for age distribution.”

65. Line 470-471: It should read, “or individuals due to the potential for ecological bias.”

66. Line 477: It should read, “variables, such as……….diseases, cannot……”

67. Lines 479-482: It should read, “could help in the identification of public policies that could limit the impact of future epidemics. Even though our study focused on the early stages of SARS-CoV-2 spread, its results allow us to………………………………should be taken to control additional waves of COVID-19 or………….”

References:

68. Make certain to not use capital letters for the first letter of each word in manuscript titles for the following references: 4, 5, 22, 33, 34.

69. Line 533: It should read, “SARS-CoV-2”.

7. PLOS authors have the option to publish the peer review history of their article (what does this mean?). If published, this will include your full peer review and any attached files.

Reviewer #2: No

---

## [Author Response · Author response to Decision Letter 2]

13 May 2021

PONE-D-20-26970R2

Spatial epidemiological study of the distribution, clustering, and risk factors associated with early COVID-19 mortality in Mexico

PLOS ONE

Dear Dr. Ramírez-Aldana,

Thank you for submitting your manuscript to PLOS ONE. After careful consideration, we feel that it has merit but does not fully meet PLOS ONE’s publication criteria as it currently stands. Therefore, we invite you to submit a revised version of the manuscript that addresses all the issues raised by the reviewer.

We look forward to receiving your revised manuscript.

Kind regards,

Agricola Odoi, BVM, MSc, PhD, FAHA, FACE

Academic Editor

PLOS ONE

Response: Thank you very much for the revision. We have modi\fed the paper attending to the comment raised by Reviewer 2. A point-by-point response is provided as an additional file. The modi\fcations have been marked in yellow in the new version of the paper.

Reviewer #2: Manuscript #: PONE-D-20-26970R2

Manuscript title: Spatial epidemiological study of the distribution, clustering, and risk factors associated with early COVID-19 mortality in Mexico

Authors: Ramirez-Aldana et al.

General comments:

The authors have done a thorough and thoughtful job in addressing reviewers’ questions and concerns in the revised manuscript and response letter. Below I have included editorial suggestions to make the text clearer and have some requests for clarification. In the future the authors need to have their work properly edited.

Response: Thank you very much for your revision and comments. We appreciate your time in reviewing and commenting on our paper. We have modified the paper attending to all your suggestions, which have greatly improved the manuscript. A point-by-point response to every raised issue is provided as an additional file.

---

## [Decision Letter · Decision Letter 3]

7 Jul 2021

Spatial epidemiological study of the distribution, clustering, and risk factors associated with early COVID-19 mortality in Mexico

PONE-D-20-26970R3

Dear Dr. Ramírez-Aldana,

We’re pleased to inform you that your manuscript has been judged scientifically suitable for publication and will be formally accepted for publication once it meets all outstanding technical requirements.

Kind regards,

Agricola Odoi, BVM, MSc, PhD, FAHA, FACE, Dipl. AVES (Hon)

Academic Editor

PLOS ONE

Additional Editor Comments (optional):

Reviewers' comments:

Reviewer's Responses to Questions

**Comments to the Author**

1. If the authors have adequately addressed your comments raised in a previous round of review and you feel that this manuscript is now acceptable for publication, you may indicate that here to bypass the “Comments to the Author” section, enter your conflict of interest statement in the “Confidential to Editor” section, and submit your "Accept" recommendation.

Reviewer #2: All comments have been addressed

2. Is the manuscript technically sound, and do the data support the conclusions?

Reviewer #2: Yes

3. Has the statistical analysis been performed appropriately and rigorously? 

Reviewer #2: Yes

4. Have the authors made all data underlying the findings in their manuscript fully available?

Reviewer #2: Yes

5. Is the manuscript presented in an intelligible fashion and written in standard English?

Reviewer #2: Yes

6. Review Comments to the Author

Reviewer #2: Manuscript ID: PONE-D-20-26970R3

Title: Spatial epidemiological study of the distribution, clustering, and risk factors associated with early COVID-19 mortality in Mexico

Authors: Ramirez-Aldana et al.

General comments: The authors’ responses to the requested revisions and questions were thoughtful and complete. There are some very minor issues/typos they may want to consider fixing, but I do not need to see a revised draft and the suggestions should be considered voluntary. After three rounds of revisions, I suspect this manuscript is in better shape than most especially considering the complexity of the authors’ analyses.

Specific comments:

1) The footnote on Table 1 would sound better as “used as the offset”.

2) Line 235: It should read, “models’ variables” if they mean the variables from the three models rather than one model.

3) Line 316: I checked on-line and it appears R uses “1/theta” which is equivalent to the overdispersion parameter “alpha” that would be familiar to SAS, STATA, and SSPS users. In any case, the extra-dispersion term in the model has a gamma distribution “gamma (1/alpha, 1/alpha)”. Consequently, I think the authors in writing (gamma <0.001, p-value=0.5) have confused the value for the LRT Chi-square for alpha=0 (or an R-equivalent) with the distribution. I would just remove the “gamma <0.001” in the brackets to avoid confusing the readers. It appears that the authors have done the correct analysis and interpreted the key result correctly.

4) Lines 317-322: The authors have indicated which interactions were tested as requested, but it only appears in the results rather than methods. I think this is acceptable and better than most manuscripts that do not indicate these effects were examined. However, in the future, I would recommend reporting which interaction effects were going to be tested first in the methods to be consistent with current reporting guidelines for observational studies.

5) Line 459: It should read, “such as....”

7. PLOS authors have the option to publish the peer review history of their article (what does this mean?). If published, this will include your full peer review and any attached files.

Reviewer #2: No

---

## [Editor Report · Acceptance letter]

9 Jul 2021

PONE-D-20-26970R3 

Spatial epidemiological study of the distribution, clustering, and risk factors associated with early COVID-19 mortality in Mexico 

Dear Dr. Ramírez-Aldana:

I'm pleased to inform you that your manuscript has been deemed suitable for publication in PLOS ONE. Congratulations! Your manuscript is now with our production department. 

Kind regards, 

on behalf of

Prof. Agricola Odoi 

Academic Editor

PLOS ONE